# Ship path following control using Event-triggered lexicographic ordering multi-objective model predictive control

Yukun Sun 1
*1 Navigation College*
*Dalian Maritime University*
Dalian, China
syk1226@dlmu.edu.cn

Yuchi Cao 1*
*1 Navigation College*
*Dalian Maritime University*
Dalian, China
18900982621@163.com

Qihe Shan 1
*1 Navigation College*
*Dalian Maritime University*
Dalian, China
shanqihe@163.com

Hanxuan Zhang 1
*1 Navigation College*
*Dalian Maritime University*
Dalian, China
zhanghanxuan1995@dlmu.edu.cn

*Abstract*—**This paper applies the lexicographic ordering method to solve the multi-objective model predictive control (MOMPC) problem for ship path following, where the multiple objectives include path convergence and speed tracking. Additionally, an event-triggered mechanism is introduced to reduce the solution frequency of the multi-objective optimization problem. Firstly, a MOMPC cost function is constructed for path convergence and speed tracking. Subsequently, the lexicographic ordering method is applied to prioritize the resolution of the path convergence issue, followed by addressing the speed tracking problem under the results obtained from solving the path convergence as constraint conditions. Moreover, to reduce the communication burden, an event-triggered mechanism is integrated into the MOMPC, effectively reducing computational load while ensuring tracking effectiveness. Finally, simulation results are presented to verify the performance of the proposed method.**

*Keywords—lexicographic ordering method, multi-objective model predictive control, path convergence, speed tracking, event-triggered mechanism*

## I. INTRODUCTION

Shipping [1], as an important mode of transportation for a country's import and export trade, has seen significant development in recent years, with the number of ships at sea increasing annually. Consequently, maritime safety issues have gained more and more attention. Therefore, maneuvering ships to follow specified routes has emerged as a pivotal area of research focus. This is not only crucial for ensuring the safety of transportation but also for improving transportation efficiency. Ship path following control [2-3] is a motion control style that manipulates ships to travel along a predetermined route. Moreover, most existing ships are underactuated [4-5], meaning they lack lateral propulsion and are equipped with only two power output devices: longitudinal thrust and steering force. The underactuated characteristics of ships not only pose challenges for actual navigation but also increase the nonlinear complexity of the ship systems. Therefore, the study of path following control for underactuated ships has profound implications for the development of the shipping industry. This research can lead to advancements in safety, efficiency, and the overall performance of maritime transportation, which are critical for the economic and strategic interests associated with sea trade.

There have been numerous research methodologies addressing the ship path following control issue, including the backstepping method [6-7], the line-of-sight (LOS) guidance method [8] inspired by sailor experience, sliding mode [9], fuzzy control [10], neural networks, and the proportional-integral-derivative (PID) approach [11]. These methods, after optimization and improvement, have been capable of achieving satisfactory tracking performance. However, due to the motion state and input constraints present during the navigation of the ship system, these methods find it challenging to address such constrained problems.

MPC possesses an exceptional capacity to manage constraints, which has led to its broad implementation across a spectrum of industries, particularly within the intricacies of advanced industrial processes. It is renowned for its capacity to optimize processes while taking into account the system's physical limitations and operational boundaries, making it a preferred control strategy in scenarios where constraints are a critical consideration. Ref [12] is based on a linearized model and uses MPC to address the steering drive limitations of maritime surface vessels in terms of amplitude and rate. Ref [13] utilizes nonlinear model predictive control (NMPC) to tackle the ship path following control problem. Traditional ship path following control often focuses solely on the convergence of the ship's path, whereas in actual navigation, maintaining a ship's travel at a given speed is equally significant. A stable speed not only ensures the safe transportation of cargo but also reduces the risk of collisions, ensuring orderly and smooth traffic in the shipping lanes.

Therefore, the MOMPC is proposed in [14], which can simultaneously consider multiple target optimization issues such as ship path convergence and speed tracking. This approach allows for a more comprehensive control strategy that addresses both the path and velocity aspects of ship navigation, enhancing the overall performance and safety of the ship's journey. In ref [15], the MOMPC method is utilized to enable fully actuated

ships to track the surge speed in addition to traditional path following.

However, MPC requires optimization calculations at every sampling moment, which can be very computationally intensive and time-consuming. Event-triggered control [16-17] is a non-periodic sampling strategy that, compared to traditional periodic sampling control, offers higher sampling efficiency and lower computational load. Therefore, it can effectively alleviate the computational burden of MPC.

In response to the aforementioned issues, this paper employs the lexicographic ordering method in MOMPC to address the ship path following control problem and introduces an event-triggered mechanism to reduce computational load and enhance computational efficiency. The merits of this paper are as follows:

(1) This paper utilizes the lexicographic ordering method to solve the MOMPC cost function by first addressing the sub-cost function with path convergence as the primary task under the conditions of control input. Subsequently, with above result as an additional constraint, the sub-cost function with speed tracking as the secondary task is solved, yielding the final optimal control sequence.

(2) The paper integrates an event-triggered strategy into MOMPC (ETMOMPC), where the optimization solution process occurs only at fixed moments periodic with the prediction horizon or at moments when the designed threshold is triggered. When the above conditions are not met, the controller uses the optimal control sequence solved from the previous moment as the input for the next moment. This approach not only reduces the frequency of solving optimization problems but also prevents resource wastage due to online computation, thereby enhancing the utilization rate of MOMPC.

The content of the remaining sections of this paper is as follows: The section 2 introduces the ship system and the reference values for ship path following. The section 3 presents the ETMOMPC algorithm and provides proof against the occurrence of Zeno phenomenon. The section 4 conducts an analysis of the algorithm's stability. The section 5 presents simulation data along with comparative simulation figures. The section 6 concludes the paper.

## II. PRELIMINARIES

### A. Ship model

This paper employs the following three degree-of-freedom (DOF) underactuated ship model. The model only takes into account the surge, sway, and yaw of the ship, and ignores heave, roll, and pitch. The specifics are as follows:

Dalian Maritime University Navigation College First-class Interdisciplinary Research Project

$$\begin{cases} \dot{x} = u\cos\psi - v\sin\psi \\ \dot{y} = u\sin\psi + v\cos\psi \\ \dot{\psi} = r \\ \dot{u} = \dfrac{M_v}{M_u}vr - \dfrac{D_{u1}}{M_u}u - \dfrac{D_{u2}}{M_u}u|u| + \dfrac{F_u}{M_u} \\ \dot{v} = -\dfrac{M_u}{M_v}ur - \dfrac{D_{v1}}{M_v}v - \dfrac{D_{v2}}{M_v}v|v| \\ \dot{r} = \dfrac{M_u - M_v}{M_r}uv - \dfrac{D_{r1}}{M_r}r - \dfrac{D_{r2}}{M_r}r|r| + \dfrac{F_r}{M_r} \end{cases} = h(\chi, \omega) \quad (1)$$

where $h(\chi, \omega)$ serves as the nonlinear function of the ship and is locally Lipschitz continuous. The Lipschitz constant [18] $L_\chi$, which is a known constant, can be calculated using the following formula:

$$\|h(\chi_1, \omega) - h(\chi_2, \omega)\| \le L_\chi \|\chi_1 - \chi_2\| \quad (2)$$

$M_u$, $M_v$ and $M_r$ are the ship's inertia term. $D_{ui}$, $D_{vi}$ and $D_{ri}(i = 1,2)$ denote damping parameters. $F_u$ represents the surge force, $F_r$ stands for the yaw moment. $\chi = [x, y, \psi, u, v, r]^{\mathrm{T}}$, $\omega = [F_u, F_r]^{\mathrm{T}}$.

### B. Reference path design

Consider the geometric path described by the following mapping relationships:

$$C = \{\mathcal{L} \in \mathbb{R}^2 \,|\, \mathcal{L} = \mathcal{L}(\vartheta), \vartheta \in [\vartheta_0, \vartheta_1]\} \quad (3)$$

The scalar $\vartheta$ is used to represent the path parameter, and the mapping relationship $\mathcal{L}$ is smooth and bounded. For the convenience of studying the path following problem, it can be represented in the form of path dynamic output.

$$\dot{\vartheta} = \ell(\vartheta, v_\vartheta), \mathcal{L} = \mathcal{L}(\vartheta) \quad (4)$$

where $v_\vartheta$ serves as the control input for the reference path.

For the ship system, each state variable has different physical significance, providing guidance for the six-dimensional reference state of the ship.

$$\mathcal{L}_C\left(\vartheta(t), \dot{\vartheta}(t)\right) =$$
$$\begin{bmatrix} \mathcal{L}_x(\vartheta(t)), \mathcal{L}_y(\vartheta(t)), \mathcal{L}_\psi(\vartheta(t)), \\ \mathcal{L}_u(\vartheta(t), \dot{\vartheta}(t)), \mathcal{L}_v(\vartheta(t)), \mathcal{L}_r(\vartheta(t), \dot{\vartheta}(t)) \end{bmatrix}^{\mathrm{T}} \quad (5)$$

where

$$\mathcal{L}_\psi(\vartheta(t)) = \operatorname{atan2}(\dot{\mathcal{L}}_y, \dot{\mathcal{L}}_x) \quad (6)$$

$$\mathcal{L}_u(\vartheta(t), \dot{\vartheta}(t)) = \sqrt{\dot{\mathcal{L}}_x^2 + \dot{\mathcal{L}}_y^2}\, \dot{\vartheta}(t) \quad (7)$$

$$\mathcal{L}_v(\vartheta(t)) = 0 \quad (8)$$

$$\mathcal{L}_r(\vartheta(t), \dot{\vartheta}(t)) = \frac{\dot{\mathcal{L}}_x \ddot{\mathcal{L}}_y - \dot{\mathcal{L}}_y \ddot{\mathcal{L}}_x}{\dot{\mathcal{L}}_x^2 + \dot{\mathcal{L}}_y^2}\, \dot{\vartheta}(t) \quad (9)$$

The primary task of this paper is path convergence:

$$\lim_{t\to\infty} \left\| \chi(t) - \mathcal{L}_C\big(\vartheta(t),\dot{\vartheta}(t)\big) \right\| = 0 \qquad (10)$$

The secondary task of this paper is to make the ship's surge speed track the reference value. First, the desired surge speed $u_d$ is provided. Then, by transforming Eq. (7), the parameters for the speed tracking task can be obtained:

$$\dot{\vartheta}_d = u_d \left( \dot{\mathcal{L}}_x{}^2 + \dot{\mathcal{L}}_y{}^2 \right)^{-\frac{1}{2}} \qquad (11)$$

Other state variables related to the speed tracking task can be represented $\mathcal{L}_d\big(\vartheta_d,\dot{\vartheta}_d\big)$

## III. THE ETMOMPC FORMULATION

### A. The MOMPC Formulation

Firstly, define the time series $\{t_k\}$ as the instance at which the optimization problem is solved at time step $k$. $(*|t_k)$ represents the prediction for subsequent moments at time $t_k$. The multi-objective ship path following control problem mainly consists of two objectives: path convergence and speed tracking. Path convergence requires the ship to follow the reference route as quickly as possible, while speed tracking requires the ship to navigate at the designed surge speed. To achieve above targets, the following multi-objective optimization problem is designed.

$$\min_{\omega_\Gamma} J(\chi_\Gamma, \omega_\Gamma) \qquad (12)$$

Subject to

$$\begin{cases} \dot{\chi}_\Gamma(s|t_k) = h_\Gamma\big(\chi_\Gamma(s|t_k),\omega_\Gamma(s|t_k)\big), s \in [t_k, t_k+N] \\ \chi_\Gamma(t_k) = \chi_\Gamma(t_k|t_k) \\ \omega_\Gamma(s|t_k) \in \Pi, s \in [t_k, t_k+N] \end{cases} \qquad (13)$$

with

$$\chi_\Gamma = [\chi \quad \vartheta]^\mathrm{T}, \omega_\Gamma = [\omega \quad v_\vartheta]^\mathrm{T},$$
$$h_\Gamma(\chi_\Gamma,\omega_\Gamma) = [h(\chi,\omega) \quad \ell(\vartheta,v_\vartheta)]^\mathrm{T} \qquad (14)$$

Eq. (13) represents the augmented system that introduces the reference quantity. $\chi_\Gamma(t_k|t_k)$ is the system state of at time $t_k$. $\Pi$ represent the constraints on the ship system's control inputs. $J(\chi_\Gamma, \omega_\Gamma) = J_1(\chi_\Gamma, \omega_\Gamma) + J_2(\chi_\Gamma, \omega_\Gamma)$ is the cost function designed for two control objectives.

The specific form of each cost function is as follows:

$$J_i\big(\chi_\Gamma(s|t_k),\omega_\Gamma(s|t_k)\big) =$$
$$L_i\big(\chi_\Gamma(s|t_k),\omega_\Gamma(s|t_k)\big) + E_i\big(\chi_\Gamma(s|t_k)\big), i = 1,2 \qquad (15)$$

where

$$L_1\big(\chi_\Gamma(s|t_k),\omega_\Gamma(s|t_k)\big) =$$
$$\int_{t_k}^{t_k+T-1} \left( \|\chi_\Gamma(\tau|t_k) - \chi_C(\tau|t_k)\|_{Q_1}^2 + \|\omega_\Gamma(\tau|t_k)\|_{R_1}^2 \right) d\tau \ (16)$$

$$L_2\big(\chi_\Gamma(s|t_k),\omega_\Gamma(s|t_k)\big) =$$
$$\int_{t_k}^{t_k+T-1} \left( \|\chi_\Gamma(\tau|t_k) - \chi_d(\tau|t_k)\|_{Q_2}^2 + \|\omega_\Gamma(\tau|t_k)\|_{R_2}^2 \right) d\tau \ (17)$$

$$E_1\big(\chi_\Gamma(s|t_k)\big) = \|\chi_\Gamma(t_k+T|t_k) - \chi_C(t_k+T|t_k)\|_{P_1}^2 \quad (18)$$

$$E_2\big(\chi_\Gamma(s|t_k)\big) = \|\chi_\Gamma(t_k+T|t_k) - \chi_d(t_k+T|t_k)\|_{P_2}^2 \quad (19)$$

where $\chi_C = \mathrm{col}\big(\mathcal{L}_C(\vartheta,\dot{\vartheta}),\vartheta\big)$ serves as the reference vector for path convergence, $\chi_d = \mathrm{col}\big(\mathcal{L}_d(\vartheta_d,\dot{\vartheta}_d),\dot{\vartheta}_d\big)$ is the reference vector speed tracking. $T$ is the prediction horizon. $Q_i$, $R_i$, and $P_i$, $i = 1,2$ denote the stage state weight matrix, control input weight matrix, and terminal state weight matrix, respectively. These matrices are all symmetric and positive definite, ensuring a robust and stable weighting of the control objectives within the optimization framework.

To address the multi-objective optimization problem, this paper applies the lexicographic ordering method [19] to handle two optimization tasks with different priority levels. First, the primary optimization objective $J_1$ is solved at each sampling instance, and then the obtained results are used as a condition to solve $J_2$.

$$J_1^*(\chi_\Gamma, \omega_\Gamma) = \min_{\omega_\Gamma}\{J_1(\chi_\Gamma, \omega_\Gamma)|(13)\} \qquad (20)$$

$$J_2^*(\chi_\Gamma, \omega_\Gamma) =$$
$$\min_{\omega_\Gamma}\{J_2(\chi_\Gamma, \omega_\Gamma)|(13), J_1(\chi_\Gamma, \omega_\Gamma) = J_1^*(\chi_\Gamma, \omega_\Gamma)\} \qquad (21)$$

We can deduce that

$$\omega_\Gamma^* = \arg\min_{\omega_\Gamma}\{J_2(\chi_\Gamma, \omega_\Gamma)|(13), J_1(\chi_\Gamma, \omega_\Gamma) = J_1^*(\chi_\Gamma, \omega_\Gamma)\} (22)$$

To avert the risk of numerical algorithm stagnation and to enhance computational efficiency, $J_1(\chi_\Gamma, \omega_\Gamma) = J_1^*(\chi_\Gamma, \omega_\Gamma)$ is frequently substituted with the following:

$$J_1(\chi_\Gamma, \omega_\Gamma) = J_1^*(\chi_\Gamma, \omega_\Gamma) + \varepsilon \qquad (23)$$

where $\varepsilon$ denotes the specified tolerance.

Under normal circumstances, path convergence and speed tracking can both be achieved physically, and due to the introduction of the tolerance $\varepsilon$, the performance of speed tracking can meet the required standards. Solving the MOMPC problem using the lexicographic ordering method not only simplifies the theoretical design but also retains the flexibility to adjust the path parameters $\vartheta(t)$.

### B. Event-triggered design.

Incorporating an event-triggered strategy into MOMPC can effectively reduce the frequency of solving the optimization problem, thereby alleviating the communication burden.

To provide a clearer description of the event-triggering mechanism, we denote the current optimization instance at time $t_k$ and express the subsequent optimization time $t_{k+1}$ as follows:

$$t_{k+1} = \min\{t_k + T, \check{t}_{k+1}\} \qquad (24)$$

The term $\check{t}_{k+1}$ represents the time at which the actual state of the ship system and the optimal state differ reaches a given trigger threshold, prompting an update in the optimization instance $\check{t}_{k+1}$. Its specific expression is $\check{t}_{k+1} = \inf\{s| \|\chi_\Gamma^*(s|t_k) - \chi_\Gamma(s|t_k)\| = \varrho\}$. $\varrho = \eta T \epsilon e^{\eta T L_\chi}$. The term inf is used to denote the infimum of a set, which is the greatest lower bound of the set. $\eta \in (0,1)$ is a constant. Due to the influence of measurement accuracy, the actual state of the ship has deviations, and here we use $\xi$ to represent the deviation, and

$\epsilon = \sup_{\xi \in \Theta} ||\xi||$ to represent the known bound of the deviation, where $\Theta$ denotes the set within which the deviation $\xi$ is encapsulated, and it is a compact set. The term sup represents the supremum of a set, which is the least upper bound of the set.

Eq. (24) delineates the specific criteria for the designed event-triggered control: triggering instances occur exclusively at fixed moments that are periodic with the prediction horizon $T$, or at moments when a given threshold is breached.

Introducing an event-triggered mechanism is crucial for ensuring the absence of Zeno behaviour. In other words, it prevents the optimization updates from being triggered an infinite number of times within a finite time span. To demonstrate the avoidance of Zeno behaviour, the following theorem is proposed.

**Theorem 1** For System (1) and Optimization Problem (12), an event-triggered mechanism (24) is designed with an upper trigger bound $T$ and a lower trigger bound $\eta T$.

**Proof** From Eq. (24), it is evident that there exists an upper bound $T$ for the event triggering. Therefore, the discussion only needs to focus on the existence of the lower bound for the event-triggering mechanism.

Consider the difference between the actual state and the optimal state of the ship:

$$\|\chi_\Gamma^*(s|t_k) - \chi_\Gamma(s|t_k)\| =$$

$$\| \int_{t_k}^s h_\Gamma\big(\chi_\Gamma^*(\tau|t_k), \omega_\Gamma^*(\tau|t_k)\big)\, d\tau + \chi_\Gamma(t_k|t_k) - \chi_\Gamma(t_k)$$

$$- \int_{t_k}^s \xi(\tau|t_k)\, d\tau - \int_{t_k}^s h_\Gamma\big(\chi_\Gamma(\tau|t_k), \omega_\Gamma^*(\tau|t_k)\big)\, d\tau \| \tag{25}$$

According to equation (13), it can be inferred that at time $t_k$, the optimal state value is equal to the actual state value. Given that $\xi$ has a known bound $\epsilon$, equation (25) can be transformed into the following inequality:

$$\|\chi_\Gamma^*(s|t_k) - \chi_\Gamma(s|t_k)\| \leq$$

$$\left\| \begin{array}{c} \int_{t_k}^s h_\Gamma\big(\chi_\Gamma^*(\tau|t_k), \omega_\Gamma^*(\tau|t_k)\big)\, d\tau - \\ \int_{t_k}^s h_\Gamma\big(\chi_\Gamma(\tau|t_k), \omega_\Gamma^*(\tau|t_k)\big)\, d\tau - \int_{t_k}^s \epsilon\, d\tau \end{array} \right\| \tag{26}$$

For equation (26), we can apply the Lipschitz condition and the triangle inequality to deduce that

$$\|\chi_\Gamma^*(s|t_k) - \chi_\Gamma(s|t_k)\| \leq$$

$$\int_{t_k}^s L_\chi \|\chi_\Gamma^*(\tau|t_k) - \chi_\Gamma(\tau|t_k)\|\, d\tau + \epsilon(s - t_k) \tag{27}$$

Equation (27) satisfies the conditions for the Gronwall-Bellman inequality, hence the following inequality can be established:

$$\|\chi_\Gamma^*(s|t_k) - \chi_\Gamma(s|t_k)\| \leq \epsilon(s - t_k)e^{\int_{t_k}^s L_\chi d\tau}$$
$$= \epsilon(s - t_k)e^{L_\chi(s - t_k)} \tag{28}$$

The left-hand side of inequality (28) corresponds to the designed event-triggering condition. According to the definition of $\check{t}_{k+1}$, $\|\chi_\Gamma^*(t_{k+1}|t_k) - \chi_\Gamma(t_{k+1}|t_k)\| = \varrho = \eta T \epsilon e^{\eta T L_\chi}$. the following inequality can be hold:

$$\eta T \epsilon e^{\eta T L_\chi} \leq \epsilon(t_{k+1} - t_k)e^{L_\chi(t_{k+1} - t_k)} \tag{29}$$

From equation (29), it can be obtained that $\eta T \leq \{t_{k+1} - t_k\}$. The proof of the existence of the lower bound in the event-triggering mechanism is complete.

## IV. STABILITY ANALYSIS

To study stability, the main idea is to ensure that the designed cost function decreases monotonically between two adjacent moments. To this end, the following assumptions are first provided.

**Assumption 1** The stage cost function $L_i$ and the terminal cost function $E_i$ are continuous, and it is satisfied that $L_i(0,0) = 0$, $E_i(0) = 0, i = 1,2$.

**Assumption 2** The set $\Sigma$ is closed set, and the set $\Pi$ representing control input constraints is a compact set. Each set includes the origin.

**Assumption 3** For the augmented ship system $h_\Gamma(\chi_\Gamma, \omega_\Gamma)$, there exists an invariant set $\Sigma$ and a local controller $K_1(\chi_\Gamma)$ such that the following inequality holds:

$$K_1(\chi_\Gamma) \in \Pi, h_\Gamma\big(\chi_\Gamma, K_1(\chi_\Gamma)\big) \in \Sigma,$$

$$E_1\big(h_\Gamma(\chi_\Gamma, K_1(\chi_\Gamma))\big) - E_1(\chi_\Gamma) + L_1\big(\chi_\Gamma, K_1(\chi_\Gamma)\big) \leq 0 \tag{30}$$

For any $\chi_\Gamma \in \Sigma$.

Next, leveraging the invariant set $\Sigma$ proposed in the assumptions, we apply the lexicographic ordering method to solve the MOMPC optimization problem, imposing a terminal constraint.

$$J_1^*(\chi_\Gamma, \omega_\Gamma) = \min_{\omega_\Gamma}\{J_1(\chi_\Gamma, \omega_\Gamma)|(13), \chi_\Gamma(t_k + T|t_k) \in \Sigma\} \tag{31}$$

$$J_2^*(\chi_\Gamma, \omega_\Gamma) =$$

$$\min_{\omega_\Gamma}\left\{ \begin{array}{c} J_2(\chi_\Gamma, \omega_\Gamma)|(13), J_1(\chi_\Gamma, \omega_\Gamma) \\ \leq J_1^*(\chi_\Gamma, \omega_\Gamma) + \varepsilon, \chi_\Gamma(t_k + T|t_k) \in \Sigma \end{array} \right\} \tag{32}$$

**Theorem 1** Assuming that Assumptions 1, 2, and 3 hold, at the moment $t_k$, for $\varepsilon = 0$, Eq. (32) is feasible. Let $\omega_\Gamma^*(t_{k-1})$ denote the optimal control input sequence obtained from solving the optimization problem at the previous moment, and $\chi_\Gamma(t_{k-1})$ represent the state of the ship system at the previous moment. When $t > t_k$, if $\varepsilon = L_1(\chi_\Gamma(t_{k-1}), \omega_\Gamma^*(t_{k-1})) - \chi_\Gamma(t_k)^T Q_\varepsilon \chi_\Gamma(t_k)$ with $0 < Q_\varepsilon < Q_1$, the system (14) is asymptotically stable.

$$\omega_\Gamma^*(t_{k-1}) = \text{col}\begin{pmatrix} \omega_\Gamma^*(t_{k-1}|t_{k-1}), \omega_\Gamma^*(t_{k-1} + 1|t_{k-1}), \dots, \\ \omega_\Gamma^*(t_{k-1} + T - 1|t_{k-1}) \end{pmatrix}$$

represent the control input sequence at the moment $t_{k-1}$, then the control input sequence at the moment $t_k$ is denoted as

$$\omega_\Gamma(t_k) = \text{col}\begin{pmatrix} \omega_\Gamma^*(t_{k-1} + 1|t_{k-1}), \dots, \\ \omega_\Gamma^*(t_{k-1} + T - 1|t_{k-1}), K_1\chi_\Gamma(t_{k-1} + T|t_{k-1}) \end{pmatrix}.$$

Next, consider the difference in the cost function $J_1$ between two adjacent moments.

$$J_1\big(\chi_\Gamma(t_k),\omega_\Gamma^*(t_k)\big) - J_1\big(\chi_\Gamma(t_{k-1}),\omega_\Gamma^*(t_{k-1})\big) \leq$$

$$J_1\big(\chi_\Gamma(t_k),\omega_\Gamma^*(t_k)\big) - J_1\big(\chi_\Gamma(t_{k-1}),\omega_\Gamma^*(t_{k-1})\big) + \varepsilon$$

$$= L_1\big(\chi_\Gamma(t_{k-1}+T|t_{k-1}), K_1\chi_\Gamma(t_{k-1}+T|t_{k-1})\big) +$$

$$E_1\left(h_\Gamma\big(\chi_\Gamma(t_{k-1}+T|t_{k-1}), K_1\chi_\Gamma(t_{k-1}+T|t_{k-1})\big)\right) -$$

$$E_1\big(\chi_\Gamma(t_{k-1}+T|t_{k-1})\big) - \chi_\Gamma(t_k)^{\mathrm{T}}Q_\varepsilon\chi_\Gamma(t_k) \qquad (33)$$

According to Assumption 3, it can be deduced that
$$J_1\big(\chi_\Gamma(t_k),\omega_\Gamma^*(t_k)\big) - J_1\big(\chi_\Gamma(t_{k-1}),\omega_\Gamma^*(t_{k-1})\big) \leq$$
$$-\chi_\Gamma(t_k)^{\mathrm{T}}Q_\varepsilon\chi_\Gamma(t_k) \leq 0 \qquad (34)$$

The cost function $J_1$ is a non-increasing function; therefore, the system (14) is asymptotically stable.

Assumption 3 is designed to identify a local feedback control law for the primary objective cost function $J_1$ and an associated terminal domain, which will ensure the ship's convergence to the reference path. Furthermore, due to the introduction of the slack variable $\varepsilon$, both path convergence and speed tracking can be achieved simultaneously, and a satisfactory speed tracking effect can be sufficiently obtained for ship path following control.

## V. SIMULATION

In this section, simulation results of the aforementioned methods are presented to provide a detailed illustration of the algorithm's tracking performance.

In this paper, the simulation step count is set to 100 with a sampling time of 0.1 seconds. The relevant parameters for the ship system are as follows: $M_u = 283.6$, $M_v = 593.2$, $M_r = 29$, $D_{u1} = 26.9$, $D_{v1} = 35.8$, $D_{r1} = 3.5$, $D_{u2} = 241.3$, $D_{v2} = 503.8$, $D_{r2} = 76.9$. The event-triggering threshold is set as $\sigma = 0.02$, the Lipschitz constant is $L_\chi = 1.2$, the prediction horizon is set to $T = 1$ second, $\epsilon = 0.1774$, $\eta = 0.1$. The initial state is designed to be $X(0) = [1,0,0,0,0,0]^{\mathrm{T}}$, $\vartheta(0) = 0$. The longitudinal thrust constraint is $-800\mathrm{N} < F_u < 800\mathrm{N}$ and the steering torque is $-600\mathrm{N}\cdot\mathrm{m} < F_u < 600\mathrm{N}\cdot\mathrm{m}$.

The reference path $\big(\mathcal{L}_x(\vartheta), \mathcal{L}_y(\vartheta)\big)$ in this paper is designed as a sine function

$$\begin{cases}\mathcal{L}_x(\vartheta) = \vartheta \\ \mathcal{L}_y(\vartheta) = \sin(\vartheta)\end{cases}$$

The desired surge speed $u_d = 1.1\mathrm{m/s}$. The weight matrices related to the cost function designed are as follows:

$$Q_1 = \mathrm{diag}(2\times10^5, 2\times10^5, 10^3, 10^3, 10^{-3}, 10^{-3}, 10^{-3})$$

$$R_1 = \mathrm{diag}(10^{-3}, 10^{-3}, 10^{-3})$$

$$P_1 = \mathrm{diag}(10^2, 10^2, 10, 80, 10^{-3}, 10^{-3}, 10^{-3})$$

$$Q_2 = \mathrm{diag}(10^4, 10^4, 10^2, 8\times10^3, 10^{-3}, 10^{-3}, 10^3)$$

$$R_2 = \mathrm{diag}(10^{-3}, 10^{-3}, 10^{-3})$$

$$P_2 = \mathrm{diag}(10^2, 10^2, 10, 10^3, 10^{-3}, 10^{-3}, 10^2)$$

Based on the parameters designed, the simulation result graph is shown below:

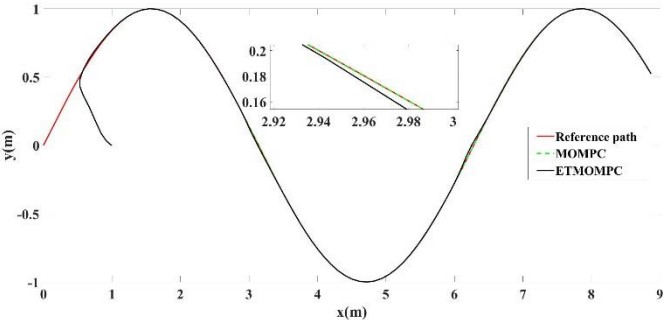

Fig. 1. *Comparison of Path Following Effects*

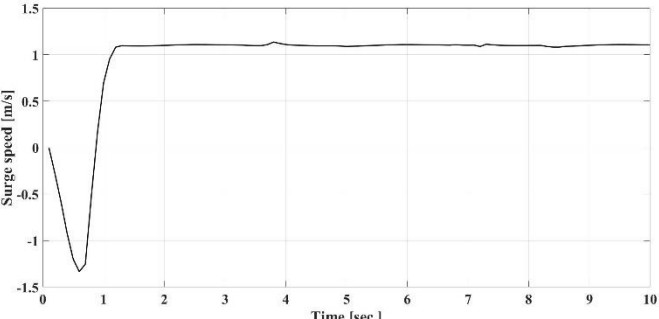

Fig. 2. *Surge Speed Tracking Performance*

Figure 1 presents a comparison of the path following effects between the MOMPC algorithm and the ETMOMPC algorithm. It can be observed that the tracking performance remains satisfactory even after the introduction of the event-triggered mechanism. Figure 2 demonstrates the tracking performance of the surge speed, showing that the ship tracks the desired surge reference speed of 1.1m/s around 1.2 seconds. The combination of Figures 1 and 2 illustrates that the algorithm can achieve both path convergence and speed tracking simultaneously.

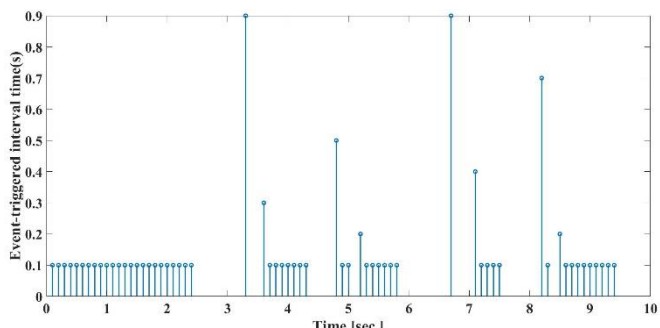

Fig. 3. *Event-triggered Interval Time*

Figure 3 illustrates the inter-event triggering intervals, and it can be observed that there is a lower bound on the intervals between consecutive triggers. This indicates that the introduction of the event-triggering mechanism does not lead to the occurrence of Zeno's phenomenon. In Figure 3, it also can be observed that after the implementation of the event-triggered mechanism, the trigger count was reduced by 39% compared to the traditional time-based sampling mechanism. This effectively reduced the computational burden.

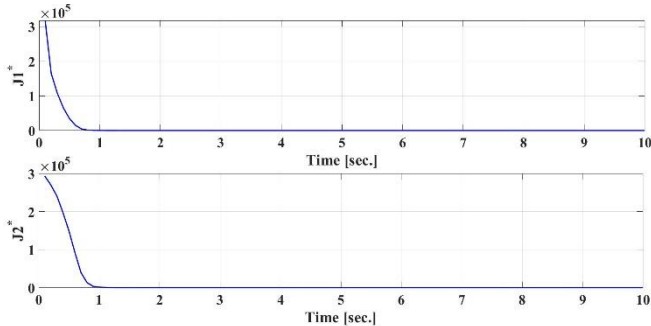

Fig. 4. *Optimal Value Cost Function*

Figure 4 presents the optimal values of the two objective functions within the cost function. It can be observed that $J_1^*$ exhibits a monotonically decreasing trend, which ensures path convergence. $J_2^*$ also shows a monotonically decreasing trend; however, the rate of decrease is comparatively slower than that of $J_1^*$, and the convergence time is longer. This indicates that the priorities for path convergence are higher than those for speed tracking, aligning with the designed lexicographic ordering method.

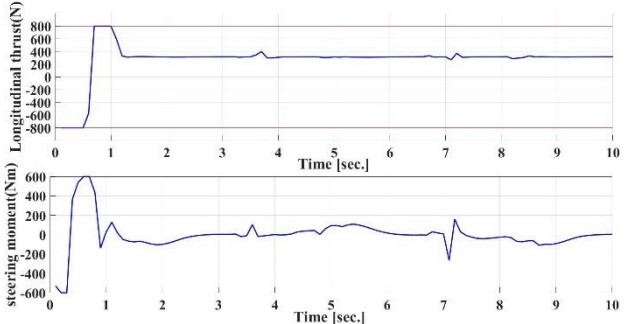

Fig. 5. *The magnitude of the control input.*

Figure 5 displays the constraint effects of the control inputs during the ship's tracking process. It can be seen that both the longitudinal thrust and the steering torque are within the designed constraint limits. This thereby demonstrates the strong constraint-handling capability of MPC.

## VI. CONCLUSION

This paper incorporates speed tracking into traditional ship path following control, forming a multi-objective tracking control problem. A cost function is constructed using MOMPC, and the optimal solution is solved using the lexicographic ordering method. Due to the high computational load and complexity inherent in MPC, an event-triggered strategy is introduced to reduce the frequency of solving optimization problems, and a proof is provided to show that Zeno behavior will not occur. Finally, the convergence of the proposed algorithm is analyzed, and simulation comparisons are presented to validate the performance of the method.

## ACKNOWLEDGMENT

This work is supported in part by Dalian Maritime University Navigation College First-class Interdisciplinary Research Project under grant number 2023JXA02.

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
