# OpenReview forum: "Ship path following control using Event-triggered lexicographic ordering multi-objective model predictive control"
_IEEE.org/ICIST/2024/Conference — IEEE ICIST 2024 Conference Submission_

### Official Review · Reviewer_tBPT · 2024-08-26
**Accept**

**Rating:** 10
**Confidence:** 5

**Review:**

This paper applies the lexicographic ordering method to solve the multi-objective model predictive control problem for ship path following, where the multiple objectives include path convergence and speed tracking. The writing and language should be improved and polished. After minor revision, it can be accepted as a conference paper.

---

### Official Review · Reviewer_3D9n · 2024-08-29
**Accept after modification**

**Rating:** 8
**Confidence:** 3

**Review:**

This paper primarily propose the lexicographic ordering method to solve the multi-objective model predictive control problem for ship path following. Generally speaking, the overall narrative of the article is complete. However, I still have some questions as follows:

1.The manuscript would benefit from further refinement in English grammar. It is advisable to review the entire article to enhance its rigor and readability.
2.The first instance of the acronym MPC in the introduction should include its full form.
3.Nomenclature should be provided for symbols introduced in the Ship model in Chapter 2, such as x, y, ψ, u, v, and r. Additionally, it would be helpful to include their practical significance.
4.The manuscript currently uses two different notations for equations, namely Eq. ( ) and equation ( ). It is advisable to review the entire text and standardize the notation to a single consistent format.
5、The images of the simulation experiments appear to be screenshots and are somewhat unclear. It is recommended to use the original, higher-resolution images to improve clarity.The first occurrence of the acronym MPC in the introduction should carry its full nomenclature.

---

### Official Review · Reviewer_8sRV · 2024-08-30
**This paper can be accepted.**

**Rating:** 7
**Confidence:** 3

**Review:**

Writing Style and Structure

Introduction: To emphasize the importance of ship path tracking control, include specific case studies or statistical data. For example, you can cite statistics on maritime accidents to highlight how effective path tracking systems can potentially reduce such incidents.

Preliminary Knowledge and Model Description:Use charts to visually represent the dynamics of a three-degree-of-freedom ship model, as well as the relationship between the reference path and the actual ship path.

Algorithm Description: When introducing the ETMOMPC formula, use pseudocode or flowcharts to clearly illustrate the steps and logic of the algorithm.

Conclusion: In addition to summarizing the research findings, suggest future work directions, such as studying the algorithm's adaptability to different sea conditions or planning real-world tests on actual ships.

Graphs and Visualizations

Simulation Results Charts: Optimize the layout and design of charts, ensuring that axis labels, legends, and titles are clear and readable. Use a consistent color scheme to differentiate between data series.

Path Tracking Performance: In charts showing path tracking performance, include visual comparisons between the reference path and the actual path, as well as quantitative representations of tracking errors.

Speed Tracking Performance: When displaying speed tracking performance, add a baseline or expected speed curve to help readers clearly see tracking performance.

Event Trigger Intervals: In charts depicting event trigger intervals, include a theoretical minimum interval line to show the relationship between actual trigger intervals and the theoretical minimum.

Control Input Magnitude: In charts showing control inputs, add constraints on the control input range to visually represent how the algorithm performs under these constraints.

Methods and Experimental Design

Algorithm Implementation Details: Provide more details about the algorithm implementation, such as the reasons for parameter selection and an analysis of the algorithm's computational complexity.

Performance Evaluation Metrics: Define clear performance evaluation metrics, such as path tracking error, speed tracking error, and smoothness of control inputs, and report these metrics in the results.

Robustness Analysis: Conduct robustness analysis to demonstrate the algorithm's performance when faced with model uncertainties and external disturbances.

---

### Decision · Program_Chairs · 2024-09-06

Accept (Oral)